# Evolution-Guided Policy Gradient in Reinforcement Learning

**Shauharda Khadka**     **Kagan Tumer**
Collaborative Robotics and Intelligent Systems Institute
Oregon State University
{khadkas,kagan.tumer}@oregonstate.edu

## Abstract

Deep Reinforcement Learning (DRL) algorithms have been successfully applied to a range of challenging control tasks. However, these methods typically suffer from three core difficulties: temporal credit assignment with sparse rewards, lack of effective exploration, and brittle convergence properties that are extremely sensitive to hyperparameters. Collectively, these challenges severely limit the applicability of these approaches to real-world problems. Evolutionary Algorithms (EAs), a class of black box optimization techniques inspired by natural evolution, are well suited to address each of these three challenges. However, EAs typically suffer from high sample complexity and struggle to solve problems that require optimization of a large number of parameters. In this paper, we introduce Evolutionary Reinforcement Learning (ERL), a hybrid algorithm that leverages the population of an EA to provide diversified data to train an RL agent, and reinserts the RL agent into the EA population periodically to inject gradient information into the EA. ERL inherits EA's ability of temporal credit assignment with a fitness metric, effective exploration with a diverse set of policies, and stability of a population-based approach and complements it with off-policy DRL's ability to leverage gradients for higher sample efficiency and faster learning. Experiments in a range of challenging continuous control benchmarks demonstrate that ERL significantly outperforms prior DRL and EA methods.

## 1  Introduction

Reinforcement learning (RL) algorithms have been successfully applied in a number of challenging domains, ranging from arcade games [35, 36], board games [49] to robotic control tasks [3, 31]. A primary driving force behind the explosion of RL in these domains is its integration with powerful non-linear function approximators like deep neural networks. This partnership with deep learning, often referred to as Deep Reinforcement Learning (DRL) has enabled RL to successfully extend to tasks with high-dimensional input and action spaces. However, widespread adoption of these techniques to real-world problems is still limited by three major challenges: temporal credit assignment with long time horizons and sparse rewards, lack of diverse exploration, and brittle convergence properties.

First, associating actions with returns when a reward is sparse (only observed after a series of actions) is difficult. This is a common occurrence in most real world domains and is often referred to as the temporal credit assignment problem [54]. Temporal Difference methods in RL use bootstrapping to address this issue but often struggle when the time horizons are long and the reward is sparse. Multi-step returns address this issue but are mostly effective in on-policy scenarios [10, 45, 46]. Off-policy multi-step learning [34, 48] have been demonstrated to be stable in recent works but require complementary correction mechanisms like importance sampling, Retrace [37, 59] and V-trace [14] which can be computationally expensive and limiting.

Secondly, RL relies on exploration to find good policies and avoid converging prematurely to local optima. Effective exploration remains a key challenge for DRL operating on high dimensional action and state spaces [41]. Many methods have been proposed to address this issue ranging from count-based exploration [38, 55], intrinsic motivation [4], curiosity [40] and variational information maximization [26]. A separate class of techniques emphasize exploration by adding noise directly to the parameter space of agents [20, 41]. However, each of these techniques either rely on complex supplementary structures or introduce sensitive parameters that are task-specific. A general strategy for exploration that is applicable across domains and learning algorithms is an active area of research.

Finally, DRL methods are notoriously sensitive to the choice of their hyperparamaters [25, 27] and often have brittle convergence properties [24]. This is particularly true for off-policy DRL that utilize a replay buffer to store and reuse past experiences [5]. The replay buffer is a vital component in enabling sample-efficient learning but pairing it with a deep non-linear function approximator leads to extremely brittle convergence properties [13, 24].

One approach well suited to address these challenges in theory is evolutionary algorithms (EA) [19, 50]. The use of a fitness metric that consolidates returns across an entire episode makes EAs indifferent to the sparsity of reward distribution and robust to long time horizons [44, 53]. EA's population-based approach also has the advantage of enabling diverse exploration, particularly when combined with explicit diversity maintenance techniques [9, 30]. Additionally, the redundancy inherent in a population also promotes robustness and stable convergence properties particularly when combined with elitism [2]. A number of recent work have used EA as an alternative to DRL with some success [8, 22, 44, 53]. However, EAs typically suffer with high sample complexity and often struggle to solve high dimensional problems that require optimization of a large number of parameters. The primary reason behind this is EA's inability to leverage powerful gradient descent methods which are at the core of the more sample-efficient DRL approaches.

In this paper, we introduce Evolutionary Reinforcement Learning (ERL), a hybrid algorithm that incorporates EA's population-based approach to generate diverse experiences to train an RL agent, and transfers the RL agent into the EA population periodically to inject gradient information into the EA. The key insight here is that an EA can be used to address the core challenges within DRL without losing out on the ability to leverage gradients for higher sample efficiency. ERL inherits EA's ability to address temporal credit assignment by its use of a fitness metric that consolidates the return of an entire episode. ERL's selection operator which operates based on this fitness exerts a selection pressure towards regions of the policy space that lead to higher episode-wide return. This process biases the state distribution towards regions that have higher long term returns. This is a form of implicit prioritization that is effective for domains with long time horizons and sparse rewards. Additionally, ERL inherits EA's population-based approach leading to redundancies that serve to stabilize the convergence properties and make the learning

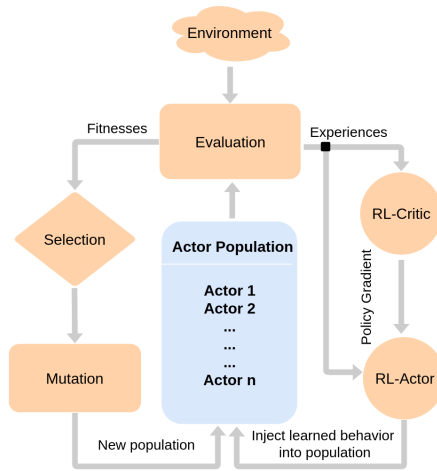

Figure 1: High level schematic of ERL highlighting the incorporation of EA's population-based learning with DRL's gradient-based optimization.

process more robust. ERL also uses the population to combine exploration in the parameter space with exploration in the action space which lead to diverse policies that explore the domain effectively.

Figure 1 illustrates ERL's double layered learning approach where the same set of data (experiences) generated by the evolutionary population is used by the reinforcement learner. The recycling of the same data enables maximal information extraction from individual experiences leading to improved sample efficiency. Experiments in a range of challenging continuous control benchmarks demonstrate that ERL significantly outperforms prior DRL and EA methods.

## 2 Background

A standard reinforcement learning setting is formalized as a Markov Decision Process (MDP) and consists of an agent interacting with an environment E over a number of discrete time steps. At each time step $t$, the agent receives a state $s_t$ and maps it to an action $a_t$ using its policy $\pi$. The agent receives a scalar reward $r_t$ and moves to the next state $s_{t+1}$. The process continues until the agent reaches a terminal state marking the end of an episode. The return $R_t = \sum_{n=1}^{\infty} \gamma^k r_{t+k}$ is the total accumulated return from time step $t$ with discount factor $\gamma \in (0, 1]$. The goal of the agent is to maximize the expected return. The state-value function $\mathcal{Q}^\pi(s, a)$ describes the expected return from state $s$ after taking action $a$ and subsequently following policy $\pi$.

### 2.1 Deep Deterministic Policy Gradient (DDPG)

Policy gradient methods frame the goal of maximizing return as the minimization of a loss function $L(\theta)$ where $\theta$ parameterizes the agent. A widely used policy gradient method is Deep Deterministic Policy Gradient (DDPG) [31], a model-free RL algorithm developed for working with continuous high dimensional actions spaces. DDPG uses an actor-critic architecture [54] maintaining a deterministic policy (actor) $\pi : \mathcal{S} \to \mathcal{A}$, and an action-value function approximation (critic) $\mathcal{Q} : \mathcal{S} \times \mathcal{A} \to \mathbb{R}$. The critic's job is to approximate the actor's action-value function $\mathcal{Q}^\pi$. Both the actor and the critic are parameterized by (deep) neural networks with $\theta^\pi$ and $\theta^\mathcal{Q}$, respectively. A separate copy of the actor $\pi'$ and critic $\mathcal{Q}'$ networks are kept as target networks for stability. These networks are updated periodically using the actor $\pi$ and critic networks $\mathcal{Q}$ modulated by a weighting parameter $\tau$.

A behavioral policy is used to explore during training. The behavioral policy is simply a noisy version of the policy: $\pi_b(s) = \pi(s) + \mathcal{N}(0, 1)$ where $\mathcal{N}$ is temporally correlated noise generated using the Ornstein-Uhlenbeck process [58]. The behavior policy is used to generate experience in the environment. After each action, the tuple $(s_t, a_t, r_t, s_{t+1})$ containing the current state, actor's action, observed reward and the next state, respectively is saved into a **cyclic replay buffer** $\mathcal{R}$. The actor and critic networks are updated by randomly sampling mini-batches from $\mathcal{R}$. The critic is trained by minimizing the loss function:

$$L = \tfrac{1}{T} \sum_i (y_i - \mathcal{Q}(s_i, a_i|\theta^\mathcal{Q}))^2 \text{ where } y_i = r_i + \gamma \mathcal{Q}'(s_{i+1}, \pi'(s_{i+1}|\theta^{\pi'})|\theta^{\mathcal{Q}'})$$

The actor is trained using the sampled policy gradient:

$$\nabla_{\theta^\pi} J \sim \tfrac{1}{T} \sum \nabla_a \mathcal{Q}(s, a|\theta^\mathcal{Q})|_{s=s_i, a=a_i} \nabla_{\theta^\pi} \pi(s|\theta^\pi)|_{s=s_i}$$

The sampled policy gradient with respect to the actor's parameters $\theta^\pi$ is computed by backpropagation through the combined actor and critic network.

### 2.2 Evolutionary Algorithm

Evolutionary algorithms (EAs) are a class of search algorithms with three primary operators: new solution generation, solution alteration, and selection [19, 50]. These operations are applied on a population of candidate solutions to continually generate novel solutions while probabilistically retaining promising ones. The selection operation is generally probabilistic, where solutions with higher fitness values have a higher probability of being selected. Assuming higher fitness values are representative of good solution quality, the overall quality of solutions will improve with each passing generation. In this work, each individual in the evolutionary algorithm defines a deep neural network. Mutation represents random perturbations to the weights (genes) of these neural networks. The evolutionary framework used here is closely related to evolving neural networks, and is often referred to as neuroevolution [18, 33, 43, 52].

## 3 Motivating Example

Consider the **standard Inverted Double Pendulum** task from OpenAI gym [6], a classic continuous control benchmark. Here, an inverted double pendulum starts in a random position, and the goal of the controller is to keep it upright. The task has a state space $\mathcal{S} = 11$ and action space $\mathcal{A} = 1$ and is a fairly easy problem to solve for most modern algorithms. Figure 2 (left) shows the comparative performance of DDPG, EA and our proposed approach: Evolutionary Reinforcement Learning

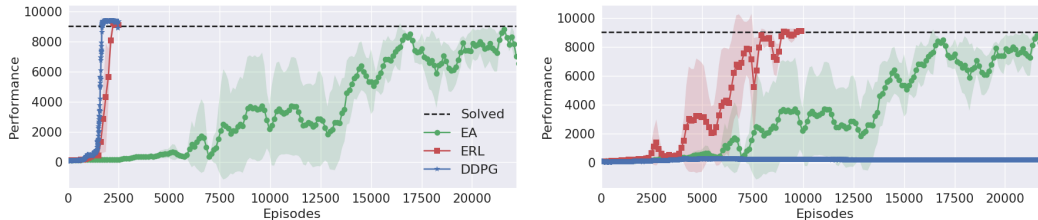

Figure 2: Comparative performance of DDPG, EA and ERL in a (left) standard and (right) hard Inverted Double Pendulum Task. DDPG solves the standard task easily but fails at the hard task. Both tasks are equivalent for the EA. ERL is able to inherit the best of DDPG and EA, successfully solving both tasks similar to EA while leveraging gradients for greater sample efficiency similar to DDPG.

(ERL), which combines the mechanisms within EA and DDPG. Unsurprisingly, both ERL and DDPG solve the task under 3000 episodes. EA solves the task eventually but is much less sample efficient, requiring approximately 22000 episodes. ERL and DDPG are able to leverage gradients that enable faster learning while EA without access to gradients is slower.

We introduce the **hard Inverted Double Pendulum** by modifying the original task such that the reward is disbursed to the controller only at the end of the episode. During an episode which can consist of up to 1000 timesteps, the controller gets a reward of 0 at each step except for the last one where the cumulative reward is given to the agent. Since the agent does not get feedback regularly on its actions but has to wait a long time to get feedback, the task poses an extremely difficult temporal credit assignment challenge.

Figure 2 (right) shows the comparative performance of the three algorithms in the **hard** Inverted Double Pendulum Task. Since EA does not use intra-episode interactions and compute fitness only based on the cumulative reward of the episode, the hard Inverted Double pendulum task is equivalent to its standard instance for an EA learner. EA retains its performance from the standard task and solves the task after 22000 episodes. DDPG on the other hand fails to solve the task entirely. The deceptiveness and sparsity of the reward where the agent has to wait up to 1000 steps to receive useful feedback signal creates a difficult temporal credit assignment problem that DDPG is unable to effectively deal with. In contrast, ERL which inherits the temporal credit assignment benefits of an encompassing fitness metric from EA is able to successfully solve the task. Even though the reward is sparse and deceptive, ERL's selection operator provides a selection pressure for policies with high episode-wide return (fitness). This biases the distribution of states stored in the buffer towards states with higher long term payoff enabling ERL to successfully solve the task. Additionally, ERL is able to leverage gradients which allows it to solve the task within 10000 episodes, much faster than the 22000 episodes required by EA. This result highlights the key capability of ERL: combining mechanisms within EA and DDPG to achieve the best of both approaches.

## 4 Evolutionary Reinforcement Learning

The principal idea behind Evolutionary Reinforcement Learning (ERL) is to incorporate EA's population-based approach to generate a diverse set of experiences while leveraging powerful gradient-based methods from DRL to learn from them. In this work, we instantiate ERL by combining a standard EA with DDPG but any off-policy reinforcement learner that utilizes an actor-critic architecture can be used.

A general flow of the ERL algorithm proceeds as follow: a population of actor networks is initialized with random weights. In addition to the population, one additional actor network (referred to as $rl_{actor}$ henceforth) is initialized alongside a critic network. The population of actors ($rl_{actor}$ excluded) are then *evaluated* in an episode of interaction with the environment. The fitness for each actor is computed as the cumulative sum of the reward that they receive over the timesteps in that episode. A *selection* operator then selects a portion of the population for survival with probability commensurate on their relative fitness scores. The actors in the population are then probabilistically *perturbed* through mutation and crossover operations to create the next generation of actors. A select portion of actors with the highest relative fitness are preserved as elites and are shielded from the mutation step.

**EA $\rightarrow$ RL:** The procedure up till now is reminiscent of a standard EA. However, unlike EA which only learns between episodes using a coarse feedback signal (fitness score), ERL additionally learns

---

**Algorithm 1** Evolutionary Reinforcement Learning

---

1: Initialize actor $\pi_{rl}$ and critic $\mathcal{Q}_{rl}$ with weights $\theta^\pi$ and $\theta^\mathcal{Q}$, respectively
2: Initialize target actor $\pi'_{rl}$ and critic $\mathcal{Q}'_{rl}$ with weights $\theta^{\pi'}$ and $\theta^{\mathcal{Q}'}$, respectively
3: Initialize a population of $k$ actors $pop_\pi$ and an empty cyclic replay buffer R
4: Define a a Ornstein-Uhlenbeck noise generator $O$ and a random number generator $r() \in [0, 1)$
5: **for** generation = 1, $\infty$ **do**
6:     **for** actor $\pi \in pop_\pi$ **do**
7:         fitness, R = Evaluate($\pi$, R, noise=None, $\xi$)
8:     **end for**
9:     Rank the population based on fitness scores
10:    Select the first $e$ actors $\pi \in pop_\pi$ as elites where $e$ = int($\psi$*k)
11:    Select $(k-e)$ actors $\pi$ from $pop_\pi$, to form Set $S$ using tournament selection with replacement
12:    **while** $|S| < (k - e)$ **do**
13:        Use crossover between a randomly sampled $\pi \in e$ and $\pi \in S$ and append to $S$
14:    **end while**
15:    **for** Actor $\pi \in$ Set $S$ **do**
16:        **if** $r() < mut_{prob}$ **then**
17:            Mutate($\theta^\pi$)
18:        **end if**
19:    **end for**
20:    _, R = Evaluate($\pi_{rl}$, R, $noise = O, \xi = 1$)
21:    Sample a random minibatch of T transitions $(s_i, a_i, r_i, s_{i+1})$ from R
22:    Compute $y_i = r_i + \gamma \mathcal{Q}'_{rl}(s_{i+1}, \pi'_{rl}(s_{i+1}|\theta^{\pi'})|\theta^{\mathcal{Q}'})$
23:    Update $\mathcal{Q}_{rl}$ by minimizing the loss: $L = \frac{1}{T} \sum_i (y_i - \mathcal{Q}_{rl}(s_i, a_i|\theta^\mathcal{Q})^2$
24:    Update $\pi_{rl}$ using the sampled policy gradient

$$\nabla_{\theta^\pi} J \sim \frac{1}{T} \sum \nabla_a \mathcal{Q}_{rl}(s, a|\theta^\mathcal{Q})|_{s=s_i, a=a_i} \nabla_{\theta^\pi} \pi(s|\theta^\pi)|_{s=s_i}$$

25:    Soft update target networks: $\theta^{\pi'} \Leftarrow \tau\theta^\pi + (1 - \tau)\theta^{\pi'}$ and $\theta^{\mathcal{Q}'} \Leftarrow \tau\theta^\mathcal{Q} + (1 - \tau)\theta^{\mathcal{Q}'}$
26:    **if** generation mod $\omega = 0$ **then**
27:        Copy the RL actor into the population: for weakest $\pi \in pop_\pi : \theta^\pi \Leftarrow \theta^{\pi_{rl}}$
28:    **end if**
29: **end for**

---

---

**Algorithm 2** Function Evaluate

---

1: **procedure** EVALUATE($\pi$, R, noise, $\xi$)
2:     $fitness = 0$
3:     **for** i = 1:$\xi$ **do**
4:         Reset environment and get initial state $s_0$
5:         **while** env is not done **do**
6:             Select action $a_t = \pi(s_t|\theta^\pi) + noise_t$
7:             Execute action $a_t$ and observe reward $r_t$ and new state $s_{t+1}$
8:             Append transition $(s_t, a_t, r_t, s_{t+1})$ to R
9:             $fitness \leftarrow fitness + r_t$ and $s = s_{t+1}$
10:        **end while**
11:    **end for**
12:    Return $\frac{fitness}{\xi}$, R
13: **end procedure**

---

from the experiences within episodes. ERL stores each actor's experiences defined by the tuple *(current state, action, next state, reward)* in its replay buffer. This is done for every interaction, at every timestep, for every episode, and for each of its actors. The critic samples a random minibatch from this replay buffer and uses it to update its parameters using gradient descent. The critic, alongside the minibatch is then used to train the $rl_{actor}$ using the sampled policy gradient. This is similar to the learning procedure for DDPG, except that the replay buffer has access to the experiences from the entire evolutionary population.

---

**Algorithm 3** Function Mutate

---
1: **procedure** MUTATE($\theta^\pi$)
2:     **for** Weight Matrix $\mathcal{M} \in \theta^\pi$ **do**
3:         **for** iteration = 1, $mut_{frac} * |\mathcal{M}|$ **do**
4:             Randomly sample indices $i$ and $j$ from $\mathcal{M}'s$ first and second axis, respectively
5:             **if** $r() < supermut_{prob}$ **then**
6:                 $\mathcal{M}[i, j] = \mathcal{M}[i, j] * \mathcal{N}(0, 100 * mut_{strength})$
7:             **else if** $r() < reset_{prob}$ **then**
8:                 $\mathcal{M}[i, j] = \mathcal{N}(0, 1)$
9:             **else**
10:               $\mathcal{M}[i, j] = \mathcal{M}[i, j] * \mathcal{N}(0, mut_{strength})$
11:             **end if**
12:         **end for**
13:     **end for**
14: **end procedure**

---

**Data Reuse:** The replay buffer is the central mechanism that enables the flow of information from the evolutionary population to the RL learner. In contrast to a standard EA which would extract the fitness metric from these experiences and disregard them immediately, ERL retains them in the buffer and engages the $rl_{actor}$ and critic to learn from them repeatedly using powerful gradient-based methods. This mechanism allows for maximal information extraction from each individual experiences leading to improved sample efficiency.

**Temporal Credit Assignment:** Since fitness scores capture episode-wide return of an individual, the selection operator exerts a strong pressure to favor individuals with higher episode-wide returns. As the buffer is populated by the experiences collected by these individuals, this process biases the state distribution towards regions that have higher episode-wide return. This serves as a form of implicit prioritization that favors experiences leading to higher long term payoffs and is effective for domains with long time horizons and sparse rewards. A RL learner that learns from this state distribution (replay buffer) is biased towards learning policies that optimizes for higher episode-wide return.

**Diverse Exploration:** A noisy version of the $rl_{actor}$ using Ornstein-Uhlenbeck [58] process is used to generate additional experiences for the replay buffer. In contrast to the population of actors which explore by noise in their *parameter space* (neural weights), the $rl_{actor}$ explores through noise in its *action space*. The two processes complement each other and collectively lead to an effective exploration strategy that is able to better explore the policy space.

**RL $\rightarrow$ EA:** Periodically, the $rl_{actor}$ network's weights are copied into the evolving population of actors, referred to as *synchronization*. The frequency of synchronization controls the flow of information from the RL learner to the evolutionary population. This is the core mechanism that enables the evolutionary framework to directly leverage the information learned through gradient descent. The process of infusing policy learned by the $rl_{actor}$ into the population also serves to stabilize learning and make it more robust to deception. If the policy learned by the $rl_{actor}$ is good, it will be selected to survive and extend its influence to the population over subsequent generations. However, if the $rl_{actor}$ is bad, it will simply be selected against and discarded. This mechanism ensures that the flow of information from the $rl_{actor}$ to the evolutionary population is constructive, and not disruptive. This is particularly relevant for domains with sparse rewards and deceptive local minima which gradient-based methods can be highly susceptible to.

Algorithm 1, 2 and 3 provide a detailed pseudocode of the ERL algorithm using DDPG as its policy gradient component. Adam [29] optimizer with gradient clipping at 10 and a learning rate of $5e^{-5}$ and $5e^{-4}$ was used for the $rl_{actor}$ and $rl_{critic}$, respectively. The size of the population $k$ was set to 10, while the elite fraction $\psi$ varied from 0.1 to 0.3 across tasks. The number of trials conducted to compute a fitness score, $\xi$ ranged from 1 to 5 across tasks. The size of the replay buffer and batch size were set to $1e^6$ and 128, respectively. The discount rate $\gamma$ and target weight $\tau$ were set to 0.99 and $1e^{-3}$, respectively. The mutation probability $mut_{prob}$ was set to 0.9 while the syncronization period $\omega$ ranged from 1 to 10 across tasks. The mutation strength $mut_{strength}$ was set to 0.1 corresponding to a 10% Gaussian noise. Finally, the mutation fraction $mut_{frac}$ was set to 0.1 while the probability from super mutation $supermut_{prob}$ and reset $resetmut_{prob}$ were set to 0.05.

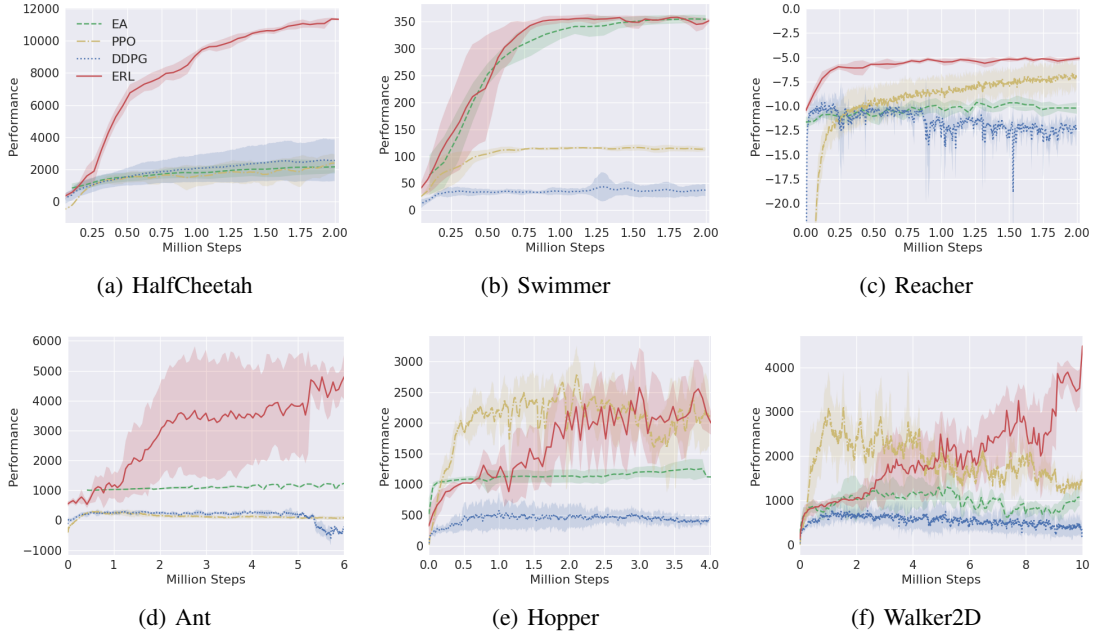

| | (a) HalfCheetah | (b) Swimmer | (c) Reacher |
|---|---|---|---|

| | (d) Ant | (e) Hopper | (f) Walker2D |
|---|---|---|---|

Figure 3: Learning curves on Mujoco-based continous control benchmarks.

## 5 Experiments

**Domain:** We evaluated the performance of ERL[1] agents on 6 continuous control tasks simulated using Mujoco [56]. These are benchmarks used widely in the field [13, 25, 53, 47] and are hosted through the OpenAI gym [6].

**Compared Baselines:** We compare the performance of ERL with a standard neuroevolutionary algorithm (EA), DDPG [31] and Proximal Policy Optimization (PPO) [47]. DDPG and PPO are state of the art deep reinforcement learning algorithms of the off-policy and and on-policy variety, respectively. PPO builds on the Trust Region Policy Optimization (TRPO) algorithm [45]. ERL is implemented using PyTorch [39] while OpenAI Baselines [11] was used to implement PPO and DDPG. The hyperparameters for both algorithms were set to match the original papers except that a larger batch size of 128 was used for DDPG which was shown to improve performance in [27].

**Methodology for Reported Metrics:** For DDPG and PPO, the actor network was periodically tested on 5 task instances without any exploratory noise. The average score was then logged as its performance. For ERL, during each training generation, the actor network with the highest fitness was selected as the champion. The champion was then tested on 5 task instances, and the average score was logged. This protocol was implemented to shield the reported metrics from any bias of the population size. Note that all scores are compared against the number of steps in the environment. Each step is defined as an instance where the agent takes an action and gets a reward back from the environment. To make the comparisons fair across single agent and population-based algorithms, all steps taken by all actors in the population are **cumulative**. For example, one episode of HalfCheetah consists of 1000 steps. For a population of 10 actors, each generation consists of evaluating the actors in an episode which would incur 10, 000 steps. We conduct five independent statistical runs with varying random seeds, and report the average with error bars logging the standard deviation.

**Results:** Figure 3 shows the comparative performance of ERL, EA, DDPG and PPO. The performances of DDPG and PPO were verified to have matched the ones reported in their original papers [31, 47]. ERL significantly outperforms DDPG across all the benchmarks. Notably, ERL is able to learn on the 3D quadruped locomotion Ant benchmark where DDPG normally fails to make any learning progress [13, 23, 24]. ERL also consistently outperforms EA across all but the Swimmer environment, where the two algorithms perform approximately equivalently. Considering

that ERL is built primarily using the subcomponents of these two algorithms, this is an important result. Additionally, ERL significantly outperforms PPO in 4 out of the 6 benchmark environments[2].

The two exceptions are Hopper and Walker2D where ERL eventually matches and exceeds PPO's performance but is less sample efficient. A common theme in these two environments is early termination of an episode if the agent falls over. Both environments also disburse a constant small reward for each step of survival to encourage the agent to hold balance. Since EA selects for episode-wide return, this setup of reward creates a strong local minimum for a policy that simply survives by balancing while staying still. This is the exact behavior EA converges to for both environments. However, while ERL is initially confined by the local minima's strong basin of attraction, it eventually breaks free from it by virtue of its RL components: temporally correlated exploration in the action space and policy gradient-based on experience batches

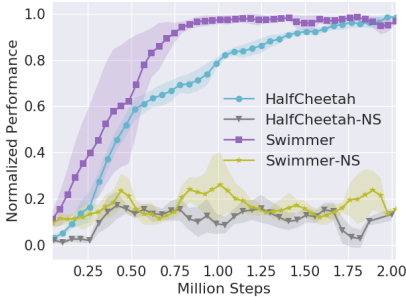

Figure 4: Ablation experiments with the selection operator removed. NS indicates ERL without the selection operator.

sampled randomly from the replay buffer. This highlights the core aspect of ERL: *incorporating the mechanisms within EA and policy gradient methods to achieve the best of both approaches.*

**Ablation Experiments**: We use an ablation experiment to test the value of the selection operator, which is the core mechanism for experience selection within ERL. Figure 4 shows the comparative results in HalfCheetah and Swimmer benchmarks. The performance for each benchmark was normalized by the best score achieved using the full ERL algorithm (Figure 3). Results demonstrate that the selection operator is a crucial part of ERL. Removing the selection operation (NS variants) lead to significant degradation in learning performance ($\sim$80%) across both benchmarks.

**Interaction between RL and EA**: To tease apart the system further, we ran some additional experiments logging whether the $rl_{actor}$ synchronized periodically within the EA population was classified as an elite, just selected, or discarded during selection (see Table 1). The results vary across tasks with Half-Cheetah's and Swimmer standing at either extremes: $rl_{actor}$ being the most and the least performant, respec-

| | Elite | Selected | Discarded |
|---|---|---|---|
| Half-Cheetah | $83.8 \pm 9.3\%$ | $14.3 \pm 9.1\%$ | $2.3 \pm 2.5\%$ |
| Swimmer | $4.0 \pm 2.8\%$ | $20.3 \pm 18.1\%$ | $76.0 \pm 20.4\%$ |
| Reacher | $68.3 \pm 9.9\%$ | $19.7 \pm 6.9\%$ | $9.0 \pm 6.9\%$ |
| Ant | $66.7 \pm 1.7\%$ | $15.0 \pm 1.4\%$ | $18.0 \pm 0.8\%$ |
| Hopper | $28.7 \pm 8.5\%$ | $33.7 \pm 4.1\%$ | $37.7 \pm 4.5\%$ |
| Walker-2d | $38.5 \pm 1.5\%$ | $39.0 \pm 1.9\%$ | $22.5 \pm 0.5\%$ |

Table 1: Selection rate for synchronized $rl_{actor}$

tively. The Swimmer's selection rate is consistent with the results in Figure 3b where EA matched ERL's performance while the RL approaches struggled. The overall distribution of selection rates suggest tight integration between the $rl_{actor}$ and the evolutionary population as the driver for successful learning. Interestingly, even for HalfCheetah which favors the $rl_{actor}$ most of the time, EA plays a critical role with 'critical interventions.' For instance, during the course of learning, the cheetah benefits from leaning forward to increase its speed which gives rise to a strong gradient in this direction. However, if the cheetah leans too much, it falls over. The gradient-based methods seem to often fall into this trap and then fail to recover as the gradient information from the new state has no guarantees of undoing the last gradient update. However, ERL with its population provides built in redundancies which selects against this deceptive trap, and eventually finds a direction for learning which avoids it. Once this deceptive trap is avoided, gradient descent can take over again in regions with better reward landscapes. These critical interventions seem to be crucial for ERL's robustness and success in the Half-Cheetah benchmark.

**Note on runtime:** On average, ERL took approximately 3% more time than DDPG to run. The majority of the added computation stem from the mutation operator, whose cost in comparison to gradient descent was minimal. Additionally, these comparisons are based on implementation of ERL without any parallelization. We anticipate a parallelized implementation of ERL to run significantly faster as corroborated by previous work in population-based approaches [8, 44, 53].

# 6 Related Work

Using evolutionary algorithms to complement reinforcement learning, and vice versa is not a new idea. Stafylopatis and Blekas combined the two using a Learning Classifier System for autonomous car control [51]. Whiteson and Stone used NEAT [52], an evolutionary algorithm that evolves both neural topology and weights to optimize function approximators representing the value function in Q-learning [60]. More recently, Colas et.al. used an evolutionary method (Goal Exploration Process) to generate diverse samples followed by a policy gradient method for fine-tuning the policy parameters [7]. From an evolutionary perspective, combining RL with EA is closely related to the idea of incorporating learning with evolution [1, 12, 57]. Fernando et al. leveraged a similar idea to tackle catastrophic forgetting in transfer learning [17] and constructing differentiable pattern producing networks capable of discovering CNN architecture automatically [16].

Recently, there has been a renewed push in the use of evolutionary algorithms to offer alternatives for (Deep) Reinforcement Learning [43]. Salimans et al. used a class of EAs called Evolutionary Strategies (ES) to achieve results competitive with DRL in Atari and robotic control tasks [44]. The authors were able to achieve significant improvements in clock time by using over a thousand parallel workers highlighting the scalability of ES approaches. Similar scalability and competitive results were demonstrated by Such et al. using a genetic algorithm with novelty search [53]. A companion paper applied novelty search [30] and Quality Diversity [9, 42] to ES to improve exploration [8]. EAs have also been widely used to optimize deep neural network architecture and hyperparmaters [28, 32]. Conversely, ideas within RL have also been used to improve EAs. Gangwani and Peng devised a genetic algorithm using imitation learning and policy gradients as crossover and mutation operator, respectively [22]. ERL provides a framework for combining these developments for potential further improved performance. For instance, the crossover and mutation operators from [22] can be readily incorporated within ERL's EA module while bias correction techniques such as [21] can be used to improve policy gradient operations within ERL.

# 7 Discussion

We presented ERL, a hybrid algorithm that leverages the population of an EA to generate diverse experiences to train an RL agent, and reinserts the RL agent into the EA population sporadically to inject gradient information into the EA. ERL inherits EA's invariance to sparse rewards with long time horizons, ability for diverse exploration, and stability of a population-based approach and complements it with DRL's ability to leverage gradients for lower sample complexity. Additionally, ERL recycles the date generated by the evolutionary population and leverages the replay buffer to learn from them repeatedly, allowing maximal information extraction from each experience leading to improved sample efficiency. Results in a range of challenging continuous control benchmarks demonstrate that ERL outperforms state-of-the-art DRL algorithms including PPO and DDPG.

From a reinforcement learning perspective, ERL can be viewed as a form of 'population-driven guide' that biases exploration towards states with higher long-term returns, promotes diversity of explored policies, and introduces redundancies for stability. From an evolutionary perspective, ERL can be viewed as a Lamarckian mechanism that enables incorporation of powerful gradient-based methods to learn at the resolution of an agent's individual experiences. In general, RL methods learn from an agent's life (individual experience tuples collected by the agent) whereas EA methods learn from an agent's death (fitness metric accumulated over a full episode). The principal mechanism behind ERL is the capability to incorporate both modes of learning: learning directly from the high resolution of individual experiences while being aligned to maximize long term return by leveraging the low resolution fitness metric.

In this paper, we used a standard EA as the evolutionary component of ERL. Incorporating more complex evolutionary sub-mechanisms is an exciting area of future work. Some examples include incorporating more informative crossover and mutation operators [22], adaptive exploration noise [20, 41], and explicit diversity maintenance techniques [8, 9, 30, 53]. Other areas of future work will incorporate implicit curriculum based techniques like Hindsight Experience Replay [3] and information theoretic techniques [15, 24] to further improve exploration. Another exciting thread of research is the extension of ERL into multiagent reinforcement learning settings where a population of agents learn and act within the same environment.

## Footnotes

[1]Code available at `https://github.com/ShawK91/erl_paper_nips18`

[2]Videos of learned policies available at https://tinyurl.com/erl-mujoco

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
