[Reviews · NeurIPS 2018]

Reviewer 1



Post discussion update: I have increased my score. The authors responded well. In particular they took to heart my concern about running more experiments to tease apart why the system is performing well. Obviously they did not run all the experiments I asked for, but I hope they consider doing even more if accepted. I would still like to emphasize that the paper is much more interesting if you remove the focus on SOTA results. Understanding why your system works well, and when it doesn't is much more likely to have a long-lasting scientific impact on the field whereas SOTA changes frequently. ===================================== This paper introduces a novel combination over evolutionary (population-based training) and value-based reinforcement learning. The work attempts to combine these approaches to improve data efficiency of evolutionary strategies while improving on the robustness, exploration and credit assignment compared with existing deep value learners. The strategy is straight-forward: add a Deep Actor-critic agent to the population. Train the RL agent in the usual way, but allow it to participate in the evolutionary activities of the population. The contribution is supported by empirical results in inverted pendulum and several harder open AI-gym games. I am leaning to accept this paper. The strengths are that it is a simple idea, it was evaluated on a small control domain first to provide intuitions to the reader and it achieves good perform in 3 of the 6 AI-gym domains shown. There are some weaknesses that limit the contribution. In particular: 1) the method does not seem to improve on the robustness and sensitivity as claimed in the motivation of using evolutionary methods in the first place. In fig 3 the new method is noisier in 2 domains and equally noisy as the competitors in the rest. 2) The paper claims SOTA in these domains compared to literature results. The baseline results reported in the paper under review in HalfCheetah, Swimmer and Hopper are worse the state of the art reported in the literature [1,2]. Either because the methods used achieved better results in [1,2] or because the SOTA in the domain was TRPO which was not reported in the paper under review. SOTA is a big claim; support it carefully. 3) There is significant over-claiming throughout the paper. E.g line 275 "best of both approaches", line 87 "maximal information extraction" 4) It is not clear why async methods like A3C were not discussed or compared against. This is critical given the SOTA claim 5) The paper did not really attempt to tease apart what was going on in the system. When evaluating how often was DDPG agent chosen for evaluation trials or did you prohibit this? What does the evaluation curve look like for the DDPG agent? That is do everything you have done, but evaluate the DDPG agent as the candidate from the population. Or allowing the population to produce the data for the DDPG and training it totally off-policy to see how well the DDPG learnings (breaking the bottom right link in fig 1). Does adding more RL agents help training? The paper is relatively clear and the is certainly original. However my concerns above highlight potential issues with quality and significance of the work. [1] https://arxiv.org/abs/1709.06560 [2] https://arxiv.org/pdf/1708.04133.pdf ++++++++++++ Ways to improve the paper that did not impact the scoring above: - you have listed some of the limitations of evolutionary methods, but I think there are much deeper things to say regarding leveraging state, reactiveness, and learning during an episode. Being honest and direct would work well for this work - the title is way to generic and vague - be precise when being critical. What does "brittle convergence properties mean" - I would say DeepRL methods are widely adopted. Consider the landscape 10 years ago. - claim V-trace is too expensive. I have no idea why - its important to note that evolutionary methods can be competitive but not better than RL methods - discussion starting on line 70 is unclear and seems not well supported by data. Say something more plain and provide data to back it up - definition of policy suggests deterministic actions - not sure what state space s = 11 means? typo - section at line 195 seems repetitive. omit

Reviewer 2



The paper proposes a simple, convincing approach to combine evolutionary reinforcement learning (RL) and more "classical" RL. The manuscript title is much too general. Evolutionary algorithms have been used for RL for a long time, and many refer to this approach as "evolutionary RL". The references ignore some basic work on evolutionary RL. One may argue that most classic work is not applied to very deep architectures with many layers. However, recurrent neural networks were considered and RNNs are typically considered to be deep architectures theses days regardless of how many layers they have. For example, see Verena Heidrich-Meisner and Christian Igel. Neuroevolution Strategies for Episodic Reinforcement Learning. Journal of Algorithms 64(4), pp. 152-168, 2009 and references therein. "The use of a fitness metric that consolidates returns across an entire episode makes EAs invariant to sparse rewards with long time horizons": I think this statement may be a bit misleading. Evolutionary RL methods are Monte Carlo RL methods. Thus, they do not exploit intermediate rewards beyond the contribution to the final return. That is, they have a conceptual drawback compared to, say, temporal difference methods in this respect. This should not be sold as an advantage for tasks without meaningful intermediate information. "The quantitative results obtained for ERL also compare very favorable to results reported for prior methods [12, 21, 22, 59], indicating that ERL achieves state-of-the-art performance on these benchmarks1." This should be quantified. The results of the alternative methods should be reported, at least in the supplement. --- I read the authors' reply. Regarding the comment in the rebuttal 'However, the low-fidelity signal that EAs use (fitness that compiles episode-wide reward into one number) does have the property of being "indifferent to the sparsity of reward distribution" [42] ...': Of course, EAs are indifferent to the reward distribution, because as Monte Carlo methods the just consider the final return. This means, they do not exploit any additional information contained in the timing/distribution of rewards. This in a drawback - and not a feature.

Reviewer 3



In this paper, the authors combine an evolutionary algorithm with a Deep RL algorithm so that the combination achieves the best of both worlds. It is successfully applied to 6 standard mujoco benchmarks. The following part of this review has been edited based on the other reviewers points, the authors rebuttal, an attempt to reproduce the results of the authors, and investigations in their code. First, investigations of the results and code revealed several points that are poorly documented in the paper. Here is a summary: * DDPG: - The hidden layer sizes in both the Actor and the Critic in the implementation are different from those mentioned in the paper. This is a problem since ddpg's performance can vary a lot with the architecture of the nets. - The Actor uses tanh non-linearities, and the Critic elu non-linearties. This is not mentioned. - The norm of the gradient of the Critic is clipped. This is not mentioned. * Evolutionary Algorithm: - Tournament selection is performed with tournament size of 3, and is done with replacement. - Individuals selected during tournament selection produce offspring with the elite through crossover until the population is filled. - The way mutations are handled is more complex that what is mentioned in the paper and involves many hyper-parameters. For each non elitist individual there is a fixed probability for his genome to mutate. If the genome mutates, then each weight of the actor can mutate in 3 different ways. "Normal" mutations involve adding a 10% Gaussian noise; "Super" mutations involve adding 100% Gaussian noise, and "Reset" mutations involve resetting the weight using a normalized center Gaussian noise. It seems that on average, around 10% of the weights of the Actor that undergoes a mutation are changed, but some parts of the code are still obscure. The whole mutation process is really messy and deserves more attention since it is supposed to account for half of the success of the method (the other half being the deep RL part). The authors should definitely put forward all the above facts in their paper as some of them play an important role in the performance of their system. About performance itself, in agreement with other reviewers, I consider that the authors should not base their paper on state-of-the-art performance claims, but rather on the simplicity and conceptual efficiency of their approach with respect to alternatives. I also agree with other reviewers that the title is too generic and that the approach should be better positionned with respect to the related literature (Baldwin effect etc.). For the rest, my opinion has not changed much so I keep the rest of the review mostly as is. The idea is simple, the execution is efficient, and the results are compelling (but see the remarks about reproducibility above). I am in favor of accepting this paper as it provides a useful contribution. Below I insist on weaknesses to help the authors improve their paper. A first point is a lack of clarity about deceptive reward signals or gradients. The authors state that the hard inverted pendulum is deceptive, but they don't explain what they mean, though this matters a lot for the main message of the paper. Indeed, ERL is supposed to improve over EA because it incorporates gradient-based information that should speed up convergence. But if this gradient-based information is deceptive, there should be no speed up, in contradiction with results of ERL in "deceptive reward" tasks. I would be glad to see a closer examination of ERL's behaviour in the context of a truly deceptive gradient task: does it reduce to the corresponding EA, i.e. does it consistently reject the deep RL policy until a policy that is close enough to the optimum has been found? In that respect, the authors should have a look at the work of Colas et al. at ICML 2018, which is closely related to theirs, and where the effect of deceptive gradients on deep RL is discussed in more details. Related to the above, details are missing about the tasks. In the standard inverted pendulum, is the action discrete in {-1,0,1} or continous in [-1,1] or something else? What is the immediate reward signal? Does it incorporate something to favor smaller actions? Is there a "success" state that may stop a trial before 1000 steps? All these details may make a difference in the results. The same kind of basic facts should also be given about the other benchmarks, and the corresponding details could be rejected into the appendices. The methodology for reported metrics has nice features, but the study could be made more rigorous with additional information: how many seeds did you use? How do you report variance? Can you say something about the statistical significance of your results? The third weakness is in the experimental results when reading Fig. 3. For instance, results of DDPG on half-cheetah seem to be lower than results published in the literature (see e.g. Henderson et al. "Deep RL that matters"). Could the authors investigate why (different implementation? Insufficient hyper-parameter tuning... ?) The fourth weakness is the related work section. First, the authors should read more about Learning Classifier Systems (LCSs), which indeed combined EAs and RL from the very start (Holland's 1975 work). The authors may read Lanzi, Butz or Wilson's papers about LCSs to take more distance about that. They should also take more distance about using EAs to obtain RL algorithms, which is now an important trend in Meta-RL at the moment. More generally, in a NIPS paper, we expect the "related work" section to provide a good overview of the many facets of the domain, which is not the case here. Finally, the last sentence "These approaches are complementary to the ERL framework and can be readily combined for potential further improved performance." is very vague and weak. What approach do the authors want to combine theirs with and how can it be "readily combined"? The authors must be much more specific here. In Section 2.2 it should be stated more clearly which EA is used exactly. There are many families, most algorithms have names, from the section we just get that the authors use an EA. Later on, more details about mutation etc. are missing. For instance, how do they initialize the networks? How many weights are perturbed during mutation? The authors may also explain why they are not using an ES, as used in most neuroevolution papers involved in the competition to deep RL (see numerous "Uber labs deep-neuroevolution papers"), why not NEAT, etc. Actually, it took me a while to realize that an ES cannot be used here, because there is no way to guarantee that the deep RL policy will comply with the probability distribution corresponding to the covariance matrix of the ES current population. Besides, ESs also perform a form of approximate gradient descent, making the advantage of using deep RL in addition less obvious. All this could be discussed. I would also be glad to see a "curriculum learning" aspect: does ERL start using EAs a lot, and accepts the deep RL policy more and more along time once the gradient is properly set? Note also that the authors may include policies obtained from imitation within their framework without harm. This would make it even richer. Finally, the paper would be stronger if performance was compared to recent state-of-the-art algorithms such as SAC (Haarnoja), TD3 (Fujimoto) or D4PG (Horgan), but I do not consider this as mandatory... More local points. l. 52: ref [19] has nothing to do with what is said, please pick a better ref. l. 195-201: this is a mere repetition of lines 76-81. Probably with a better view of related work, you can get a richer intro and avoid this repetition. p.2: "However, each of these techniques either rely on complex supplementary structures or introduce sensitive parameters that are task-specific": actually, the author's method also introduce supplementary structures (the EA/the deep RL part), and they also have task-specific parameters... The authors keep a standard replay buffer of 1e^6 samples. But the more actors, the faster the former content is washed out. Did the authors consider increasing the size of the replay buffer with the number of agents? Any thoughts or results about this would be welcome. l.202 sq. (diverse exploration): the fact that combining parameter noise exploration with action exploration "collectively lead to an effective exploration strategy" is not obvious and needs to be empirically supported, for instance using ablative studies about both forms of exploration. typos: I would always write "(whatever)-based" rather than "(whatever) based" (e;g. gradient-based). Google seems to agree with me. l. 57: suffer with => from l. 108: These network(s) l. 141: gradients that enable(s) l. 202: is used (to) generate l. 217: minimas => minima (latin plural of minimum) l. 258: favorable => favorably? l.266: a strong local minima => minimum (see above ;)) In ref [55] baldwin => Baldwin.